# Multipurpose Processing Additives for Silica/Rubber Composites: Synthesis, Characterization, and Application

**DOI:** 10.3390/polym13213608

**Published:** 2021-10-20

**Authors:** Arpan Datta Sarma, Carlos Eloy Federico, Frida Nzulu, Marc Weydert, Pierre Verge, Daniel Frederick Schmidt

**Affiliations:** 1Department of Materials Research and Technology (MRT), Institute of Science and Technology, L-4362 Esch-sur-Alzette, Luxembourg or iamarpan1990@gmail.com (A.D.S.); carloseloy.federico@list.lu (C.E.F.); pierre.verge@list.lu (P.V.); 2The Faculty of Science, Technology and Medicine, University of Luxembourg, L-4365 Esch-sur-Alzette, Luxembourg; 3Goodyear Innovation Center Luxembourg, L-7750 Colmar-Berg, Luxembourg; frida_nzulu@goodyear.com (F.N.); marc_weydert@goodyear.com (M.W.)

**Keywords:** soybean oil, amination, bio-based, processing aids, rubber compounds, silica particle

## Abstract

Processing additives are a special group of chemicals included in rubber formulations to facilitate the flowability of the resultant compounds. Their addition generally affects the cured properties of the subsequent rubber composites, and fine-tuning of the compound formulation is therefore required. In this work, an attempt has been made to address this issue through the preparation of new bio-based processing additives capable of promoting the mixing of the rubber compound while at the same time enhancing mechanical properties following curing. A significant decrease in the mixing energy at the first stage of mixing (~10%) has been observed by substituting only a small percentage of the conventional petroleum-derived process oil with aminated epoxidized soybean oil. Concomitantly, it is found that this aminated epoxidized soybean oil promotes rubber curing and increases the tensile strength of the final composite by ~20% compared to the control.

## 1. Introduction

Since the mid-20th century, composite materials have been of significant interest to the scientific and industrial communities [1]. The unique viscoelastic properties and low density of elastomers have made them the first choice as matrix materials in high-performance composites where high stiffness is not desired but energy dissipation is essential during end-use, e.g., tires [2]. Despite their unique properties, elastomers are seldom used on their own due to their lack of mechanical strength and durability, which makes the incorporation of reinforcing fillers essential. Nowadays, precipitated silica is preferred over carbon black to impart low rolling resistance and high wet traction for tire applications [3]. However, the incorporation of large amounts of inorganic fillers (>50–70 parts per hundred parts of rubber) limits the processability of the resultant rubber composites. This necessitates the fine-tuning of the compound formulation and/or the mixing sequence, and may additionally require the use of special mixing equipment. Of these options, the modification of the compound formulation is typically favored as the easiest solution to implement. In particular, processing additives are often introduced into the formulation for this purpose. These additives aim to enhance the viscous flow of the rubber during the different stages of processing. However, they can adversely impact the utility of the cured material by reducing its strength and modulus. Thus, a multipurpose additive that can simultaneously improve the processing efficiency of the uncured compound and the strength of the cured material would be of great interest to the rubber industry. As of the writing of this article, however, the authors are unaware of any studies reporting additives able to improve both the processing and the mechanical strength of silica-filled rubber compounds. In parallel with the desire for increased performance, the substitution of petroleum-based processing additives by bio-based alternatives is attracting increasing attention as a means of enhancing the sustainability of the resulting rubber composites these days [4]. The use of raw vegetable oils as processing additives in silica-filled rubber compounds is well documented [5,6,7,8,9] and shows real potential. These benefits can be further enhanced via suitable chemical modifications [7]. The epoxidation of the unsaturations found in vegetable oils is one of the most well-studied approaches both in academia [10,11,12] and industry [13,14,15].

Amines are known to be excellent cure accelerators in the context of sulfur-cured rubbers, and can additionally provide increased crosslink densities and enhanced mechanical properties [16]. The amination of epoxidized vegetable oils may be a good approach in this context to combine the processing capability of epoxidized vegetable oils [12] with the accelerating effect of amines. As long as triglycerides such as soybean oil are concerned, the risk of amidation of the ester bonds requires specific attention; the focus of this effort has therefore been on synthetic approaches that avoid amidation as much as possible. In this context, the amination of epoxidized soybean oil (ESO) has been carried out with different secondary amines, namely diethylamine (DEA), diisopropylamine (DIPA), diisobutylamine (DIBA), bis(2-methoxyethyl)amine (bMOEA), bis(2-hydroxypropyl)amine (bHPA), dicyclohexylamine (DCHA), pyrrolidine (Py), piperidine (Pip), and azepane (Az). The amines were chosen in order to provide varying levels of bulkiness/steric hindrance (with DCHA, DIPA, and DIBA providing the greatest level of steric hindrance) and *pK_a_* values (with Py and Pip possessing *pK_a_* values of ~11.3, followed by DEA, DIPA, and DIBA, possessing *pK_a_* values of ~11, and bMOEA, possessing a lower *pK_a_* value of ~9) [17,18]. It is assumed here that the *pK_a_* value of the parent amine will be reflected in the *pK_a_* value of the modified oil, with more alkaline amines yielding more alkaline aminated epoxidized soybean oils. This difference in alkalinity, in turn, is expected to change the catalytic ability of the aminated epoxidized soybean oil and the cured properties of the rubber compounds. With that in mind, the aminated epoxidized soybean oils thus produced were incorporated at different concentrations into model passenger car tire tread compounds [19,20], which were then cured and their mechanical properties (e.g., quasi-static mechanical properties) were investigated. The effects of the modified oils on the cure kinetics of the rubber compound were followed using a moving die rheometer (MDR), while the crosslink density was assessed by measuring equilibrium solvent swelling. Additionally, the effects of the modified oils on the dispersion of silica were monitored using micro-computed X-ray tomography. Aminated epoxidized soybean oils were found to increase crosslink density and enhance the mechanical properties of the cured compounds when incorporated into the formulation. An in-depth study of the aforementioned properties of compounds containing DEA-modified epoxidized soybean oil is first presented in order to reveal representative composition–processing–properties relations for this family of materials. Then, specific comparisons involving epoxidized soybean oil aminated using other amines are presented in order to better generalize the behavior of these systems.

## 2. Materials and Methods

### 2.1. Materials

Styrene–butadiene rubber (SBR, Sprintan SLR 4602, Trinseo, Tessenderlo, Belgium) with a styrene content of 21 mol% and a vinyl content of 50 mol% (based on the total rubber hydrocarbon content) was used for the present study along with neodymium catalyzed butadiene rubber (BR, Budene 1223, Goodyear Chemical, Beaumont, TX, USA). The Mooney viscosities (ML (1 + 4) at 100 °C) of the chosen grades of SBR and BR were 63.0 and 55.0, respectively. Epoxidized soybean oil (Makplast SN, Makwell, Mumbai, India) with an oxirane oxygen content of ~6.1% (~4.4 epoxies per triglyceride ≅ ~100% unsaturations converted to epoxies) was used to synthesize the aminated epoxidized soybean oils described here. All the amines were at least 98% pure and were procured from Sigma-Aldrich (Overijse, Belgium). Zinc chloride (97%) was purchased from Carl Roth (Karlsruhe, Germany) and dried overnight (~12–16 h) in an oven (Memmert, Buechenbach, Germany) at 150 °C before each use. Sodium bicarbonate (assay > 99%) was purchased from Merck Millipore SA (Brussels, Belgium). Dichloromethane (99%), ethanol (95%), and magnesium sulfate heptahydrate (99.5%) were procured from Acros Organics (Geel, Belgium). Deuterated chloroform (99.96%) used for NMR analysis was procured from Sigma Aldrich (Overijse, Belgium). All rubber chemicals were of commercial grade. Treated distillate aromatic extract (TDAE oil) was procured from PSP Specialties (Bangkok, Thailand). Zinc oxide (Zinc Oxide Z05) was procured from Hepşen Kimya (İstanbul, Turkey). Stearic acid (Stearic Acid) was procured from Godrej Industries (Maharashtra, India). Silica (Zeosil Premium 200MP, specific surface area ~200 m^2^/g by the CTAB technique, ASTM D6845) was procured from Solvay (Brussels, Belgium). *N*-(1,3-Dimethylbutyl)-*N*′-phenyl-*p*-phenylenediamine (Santoflex 6PPD) was procured from Eastman (Gent, Belgium). Bis(3-triethoxysilylpropyl)disulfide (Si 266) was procured from Evonik Industries (Essen, Germany). Sulfur (ground sulfur) was procured from Grupa Azoty Siarkopol (Grzybów, Poland). 2-Mercaptobenzothiazole (Vulkacit Merkapto/C) was procured from Lanxess (Köln, Germany). Diphenyl guanidine (Denax DPG) was procured from Draslovka (Kolin, Czechia). Finally, *N*-cyclohexyl-2-benzothiazolesulfenamide (Accelerator CZ) was procured from Kemai Chemical (Darmstadt, Germany). Anhydrous magnesium sulfate was prepared in-house by heating a saturated aqueous solution of magnesium sulfate in a stainless steel pan (~500 mL) using a hot plate (Severin 1500 watt) set to maximum heating. Following evaporation of all liquid water, heating was continued for another ~6–8 h (T > 200 °C) to obtain an anhydrous cake. The cake was allowed to cool down to ~70 °C, broken into ~1 cm sized pieces, then stored in a sealed glass jar before use.

### 2.2. Synthesis of Aminated Epoxidized Soybean Oil

The synthesis of aminated epoxidized soybean oil was performed following the work of Biswas et al. [21] with some adjustments of the procedure. Specifically, the reaction atmosphere and the purification process were modified. The complete procedure including adaptations was as follows: epoxidized soybean oil (1.00 mmol, 0.95 g) was mixed with the selected amine (7.15 mmol) in a Schlenk tube (Borosilicate 3.3, Rettberg, Göttingen, Germany), and then 0.15 g (1.10 mmol) of dried zinc chloride was added. The Schlenk tube was then closed with a rubber septum (Carl Roth, Karlsruhe, Germany) and the flow of argon (≥99.9999%; Alphagaz^TM^ 2, Air Liquide, Düsseldorf, Germany) with a delivery pressure of ~0.2 bar was initiated to produce an inert atmosphere. After flowing argon for ~15 min, the stopcock of the Schlenk tube was closed, keeping an overpressure of ~0.25 bar inside the Schlenk line, and the argon flow was stopped. The temperature of the solution was then increased and kept constant at 80 °C while stirring [21]. A magnetic stirrer (Hei-tec, Heidolph, Schwabach, Germany) equipped with a Pt 1000 temperature sensor was used for heating the reaction mixture. Note that the temperature chosen for the reaction exceeded the boiling point of DEA (~56 °C) and approached the boiling point of DIPA (~84 °C). An overpressure (~0.25 bar) of argon was introduced to keep DEA in liquid form under the reaction conditions, with the same procedures applied for all other amines for the sake of consistency. Furthermore, special care was taken to choose glassware, heating, and temperature control systems in order to ensure safety at these elevated temperatures and pressures.

Changes in the appearance and physical characteristics of the reaction mixture were observed during the course of the reaction. In particular, a continuous increase in the viscosity was noted as a function of reaction time, coupled with a change in the appearance of the reaction mixture from heterogeneous, translucent, and colorless to homogenous, opaque, and brown. After completion of the reaction, dichloromethane (~20 mL) was used to solubilize the product, and then the solution was transferred to a separatory funnel. The product was washed three times with 5 mL of 5 wt% sodium bicarbonate solution and then three times with 5 mL of distilled water (50–65 µS/cm) to remove the zinc chloride. The organic layer was then dried with anhydrous magnesium sulfate and filtered. The dried organic layer (solution A) was then treated to remove the unreacted amines and the remaining solvent with the procedures depending on the amine type.

In the case of oils modified with the more volatile amines (DEA, DIPA, Py, and Pip), unreacted amines and the solvent were removed via a rotary evaporator (Rotavapor R-300, Buchi, Flawil, Switzerland), and then under a dynamic vacuum of 0.5–4 mbar at 60 °C for 6–8 h. An oil pump (Edwards RV3, West Sussex, UK) was used to generate the vacuum and a liquid nitrogen-cooled cold trap was used to prevent the solvent vapors from entering the pump. In the case of oils modified with the less volatile amines (DIBA, bMOEA, bHPA, DCHA, and Az), the dried organic layer (solution A) was concentrated (up to ~5 mL) using a rotary evaporator and introduced to a silica gel packed column taking dichloromethane as the mobile phase. In the beginning, both the product and the unreacted amines were adsorbed by the column material. Ethyl acetate (~50 mL) was then introduced to the column to recover the product without any trace of amines. The solvent was then removed using the same rotary evaporator protocol as stated above. The detailed characterization of the synthesized materials is reported in the Appendix A (Appendix A; Appendix A), while the resulting inferences are discussed here as a means of explaining materials selection. With their higher *pKa* values, the heterocyclic amines (Py, Pip, Az) were found to react with the ester linkages, resulting in the breakdown of the triglyceride structure. Attack on the triglyceride esters was also observed in the case of bHPA due to the presence of free hydroxyls in the structure of the amine. On the other extreme, no reaction at all was observed with DCHA; this was ascribed to the very high level of steric hindrance of the amine group in this compound. In contrast, the modification reactions involving DEA, DIPA, DIBA, and bMOEA were successful and yielded sufficient material to enable further study. Nonetheless, the kinetics of the modification reactions involving the aforementioned amines varied substantially, yielding significant differences in the extent of grafting at a given reaction time. To facilitate comparisons between these different modification chemistries, therefore, the reaction times were adjusted to produce aminated epoxidized oils with ~1 amine group per molecule in all cases. The resultant aminated epoxidized soybean oils were incorporated into rubber compounds to form the basis for the rest of the work described here. The modified oils are named with the following nomenclature: A_x_-ESO, where “A” and “x” are the type and number of amine grafts per triglyceride, respectively. The overall approach used for the successful amination of epoxidized soybean oil with the aforementioned secondary amines is summarized schematically in Figure 1.

### 2.3. Incorporation of Aminated Epoxidized Soybean Oils in Silica-Loaded Rubber Formulations

A HAAKE PolyLab QC internal mixer (Thermo Scientific, Waltham, MA, USA) with a Rheomix 600 mixing chamber having a free volume of 85 cc was used for the mixing of all rubber compounds. The instrument was equipped with two cam-type rotors. The fill factor was kept constant at 0.75 for all mixing operations. Each mixing operation was subdivided into three steps. The mixing protocol is provided in the Appendix A (Appendix A). The initial two steps were referred to as non-productive steps (abbreviated as NP; without the addition of any curatives), while the final step was referred to as the productive step (abbreviated as PR; during which curatives were added). While the initial temperature setpoint for the non-productive mixing steps (NP_1 and NP_2) was 80 °C, the actual temperature was observed to increase up to 145 ± 5 °C during both NP_1 and NP_2 due to shear heating. During the productive stage, an initial temperature setpoint of 60 °C was used, whereas the actual temperature was observed to increase to 80 ± 5 °C during mixing. Compositional information for all formulations is gathered and tabulated in Table 1, while Table 2 shows the sequence of mixing. All rubber compounds are named based on the type of oil and its quantity in the compound, with the following nomenclature: [A_x_-ESO]_content of experimental oil_. For example, [DEA_1_-ESO]_4.5_ stands for a compound containing 4.5 phr of DEA_1_-ESO (and 21.5 phr of TDAE oil). For the volume fraction calculations, the specific gravity of the aminated epoxidized soybean oil has been assumed to be similar to that of epoxidized soybean oil (0.95, according to the supplier’s datasheet).

### 2.4. Cure Kinetics and Vulcanization of Rubber Compounds

A moving die rheometer (MDR 2000, Alpha Technologies, Heilbronn, Germany) was used to monitor the cure kinetics of the compounded rubber samples. Experiments were performed at 150 °C with a 0.5° (~7% strain) oscillation amplitude at a measurement frequency of 1.667 Hz. The equipment was preheated to 150 °C for 20–30 min to reach thermal equilibrium before loading the sample and initiating the measurement. Scorch time and optimum cure time (*t_90_*) were calculated from the curves generated by the MDR measurements. By definition, the scorch time is the time required to initiate the curing process, generally measured by the time required for a two-unit (dN.m) increase in the torque modulus from its minimum value. The time required to reach 90% of the maximum observed torque modulus is defined here as the optimum curing time and calculated using the following equation:(1)$$t_{90\;}=the\;time\;required\;to\;reach\;\frac{90}{100}\left(M_{H}\;-\;M_{L}\right)\;+\;M_{L}$$
where *M_H_* and *M_L_* are the highest and lowest recorded torque moduli during the MDR experiment.

Following cure kinetics studies via MDR, specimens of each compound were cured for the optimum cure time (*t_90_*) using a lab-scale hot press (Labtech Engineering, Praksa, Thailand) at 150 °C and a pressure of 150 bar. A stainless steel mold with a picture frame geometry (150 × 150 × 2 mm^3^) was used to shape the material during the process of vulcanization. Based on the mold geometry and the specific gravity of the material, a suitable amount (~53 g) of green compound was added, which produced a sheet of cured rubber material with a thickness of ~1.9–2.1 mm following vulcanization.

### 2.5. Specific Equilibrium Swelling of Cured Rubber Materials

Specific equilibrium swelling of cured rubber samples was determined by immersing them in toluene, consistent with the recommendations of ASTM D3616-95 for SBR rubbers [22]. Rubber samples of cubic geometry and weight of 0.1 ± 0.02 g were immersed in 20 mL of toluene. The samples were kept in the solvent for three days and weighed daily to confirm the state of swelling. After reaching equilibrium (as judged by a constant swollen mass), the weight of the swollen samples was noted (*m_s_*). The swollen samples were then dried in an oven (Memmert, Buechenbach, Germany) at 60 °C for 6–8 h. The dry mass of the rubber samples was noted (*m_d_*). The equilibrium specific swelling of the rubber sample was determined using Equation (2):(2)$$specific\;swelling=\frac{m_{s}-m_{d}}{m_{d}}$$

### 2.6. Mechanical Properties of Rubber Compounds

Dumbbell-shaped samples were punched out from the cured rubber sheets using an ISO 37-2 die. The samples were conditioned at 23 ± 5 °C for 8 to 16 h. The conditioned samples were then subjected to mechanical testing. An Instron 5967 universal testing machine (Darmstadt, Germany) was used to measure the tensile properties of the samples at 23 °C. An extension rate of 200 mm/min was applied according to ISO 37 during all the experiments. To generate statistically valid results, three samples per formulation were tested.

Strain sweep studies (in shear mode) of cured rubber samples were performed using a rubber process analyzer (RPA 2000, Alpha Technologies, Heilbronn, Germany). Green rubber samples were introduced into the instrument and cured for 10 min at 170 °C prior to any strain sweep studies. The samples were then cooled down to test the temperature of 100 °C and the strain sweep studies were performed over a strain range of 1–10% at a frequency of 1 Hz.

### 2.7. Micro-Computed X-ray Tomography (µCT) to Assess Microscale Dispersion of Silica

The µCT technique was used to study the three-dimensional morphology of silica agglomerates [23,24], using an EasyTom 160 µCT (RX Solutions, Chavanod, France) with an X-ray power setting of 50 kV and 120 µA. A frame rate of 5 frames per second was used and the samples were rotated 360° with angular steps of 0.25° during the scan. The voxel size was kept constant at 2 µm. The sample volume was reconstructed using the X-Act software package (RX Solutions, Chavanod, France), and the Avizo software package (Thermo Fisher Scientific, Waltham, MA, US) was used for image treatment and analysis. A median filter was used to de-noise the reconstructed images, which allows the segmentation of heterogeneities by thresholding the greyscale intensity histogram. Small objects (<2.5 times the voxel size) were removed to reduce uncertainties in the measurements. The images were then statistically characterized for various morphological parameters, including the equivalent diameter ($D_{eq}$) and skewness (*G_1_*) of the distribution of equivalent diameters of filler agglomerates. The skewness (*G_1_*) is a measure of the asymmetry of the frequency distribution of the equivalent diameters of the filler agglomerates and represents a means of assessing how uniform the composite morphology is over the scale of analyzed length. Prior work has shown that skewness can serve as a means of quantifying dispersion levels and composite performance [25,26,27]. The mathematical expression and exact definitions of all the quantities are summarized in the Appendix A (Equation (S1) and (S2)).

### 2.8. Data Analysis

Data analysis, plotting, and curve fitting was performed using OriginPro 2019b (Northampton, MA, US), which is the source of the reported *r^2^* values. All the fits were error-weighted using experimental uncertainties, which were obtained by repeating the experiments.

## 3. Results

### 3.1. Effect of DEA_1_-ESO on the Processability of Silica-Filled Rubber System

According to ASTM D1566-19, a processing additive can be defined as “a compounding material that improves the processability of a polymeric compound by reducing nerve, providing better dispersion of dry material, increasing the extension rate, reducing power consumption during mixing, producing smoother surfaces on calendered and extruded products and improving knitting to name a few examples” [28]. Here, we are focusing on the type of processing additives capable of reducing the mixing energy. The mixing energy is obtained directly from the HAAKE PolyLab QC internal mixer and was recorded for all the stages of mixing. As they are added in the NP_1 stage, the effect of the aminated epoxidized soybean oils on mixing energy is clearest at this point; as such, the cumulative mixing energy during the NP_1 stage has been taken here as the most relevant means of assessing changes in processability as a result of the use of different processing additives. The relevant values are presented in Figure 2 as a function of the volume percentage of DEA_1_-ESO in the formulation, while the total mixing energies including the cumulative mixing energies for all the stages are shown in the Appendix A (Appendix A). A gradual decrease in mixing energy is observed in Figure 2 as the DEA_1_-ESO content in the formulation increases, in line with our previous efforts involving epoxidized soybean oil [12]. Enhanced mixing efficiency, associated with reduced mixing energy, may be ascribed to the increased polarity of aminated epoxidized soybean oils. It is hypothesized that the polar groups of aminated epoxidized soybean oils (grafted amines, generated hydroxyls, remaining epoxies) interact with the surface of the silica particles; in contrast, without such groups in the structure, no such interaction is possible for the petroleum oils. Such enhanced interactions would, in turn, be expected to lead to hydrophobization of the silica surface, thus promoting the incorporation of the filler in the matrix. The effective hydrophobization of silica can account for the enhanced mixing efficiencies of the rubber compounds [20,29,30,31]. It is noteworthy that, above a certain concentration of DEA_1_-ESO, a plateau in mixing energy appears to be developing. This may be explained by the saturation of the silica surface with modified oil, such that no additional interactions with the silica surface can occur even when additional modified oil is added. Based on these results, this effort focused on the partial replacement of TDAE oil by DEA_1_-ESO instead of its complete replacement.

Additional work has been carried out to further elucidate the nature of these interactions through the development and application of advanced NMR techniques, and has been reported in a separate submission [32].

Here, it is worth mentioning that during the present study we have formulated rubber compounds with aminated epoxidized soybean oils possessing varying levels of amination. However, we have not observed any significant change in processing rheology depending on the number of grafted amines per triglyceride (Appendix A).

### 3.2. Effects of DEA_1_-ESO on Cure Rate and Cure State of Silica-Filled Rubber Compounds 

It is important to study the cure kinetics before any bulk vulcanization of rubber compounds to understand what is required to achieve consistent curing. For this purpose, all formulations in Table 1 were subjected to MDR analysis at 150 °C. The MDR plots of some representative samples are attached in the Appendix A (Appendix A). Values of scorch time and of the magnitude of the difference between the maximum (*M_H_*) and minimum torque (*M_L_*) (the delta torque, *∆M* = *M_H_ – M_L_*) observed during MDR analysis are shown in Figure 3a,b as a function of DEA_1_-ESO content. From Figure 3a, scorch time is observed to decrease linearly as a function of the volume percent of DEA_1_-ESO present in the formulation. The change in cure kinetics can be attributed to the presence of grafted amines, generated hydroxyls, and remaining epoxies in the structure of DEA_1_-ESO. A similar decrease in scorch time was observed in our previous study [12] thanks to the presence of epoxy groups. However, the changes observed here are much more significant thanks to the presence of amine groups. Amines are known for their catalytic action during the vulcanization reaction [16,33] and hence can affect both the rate and state of cure. In addition to the remaining epoxy groups [12], the hydroxyls generated via the epoxy-amine reaction can also interact with the silica, providing another means of reducing the adsorption of curatives on the silica surface [34,35]. Hence, the reduction in scorch time can be explained by the combined effect of all the polar groups present in the structure of DEA_1_-ESO.

From Figure 3b, the overall increase in the delta torque for all the samples containing DEA_1_-ESO can be attributed to the higher state of cure due to the presence of grafted amine groups. To support this supposition, the specific swelling of the rubber compound has also been studied. The specific swelling values of all the rubber compounds containing DEA_1_-ESO are shown in Figure 4. The small but statistically significant decrease observed in specific swelling implies an increase in crosslink density, supporting the supposition that the extent of cure increases with increasing DEA_1_-ESO content in the rubber formulation. 

### 3.3. Quasistatic Mechanical Properties of Silica-Filled Rubber System Compounded with DEA_1_-ESO

As shown above, the presence of aminated epoxidized soybean oils has been shown to enhance processability in the context of the silica-filled SBR compounds. Here, the effect of these same additives on the quasi-static mechanical properties of the respective cured compounds is shown (Figure 5). In particular, a clear improvement in tensile strength with the incorporation of aminated epoxidized soybean oil is observed. The stress vs. strain curves for the representative samples are provided in the Appendix A (Appendix A). It is important to note that the tensile strength increases without significant loss in elongation at break. This lack of a parallel decrease in elongation at break represents a potential advantage in the use of aminated epoxidized soybean oils as processing additives.

### 3.4. Dynamic Mechanical Properties of Silica-Filled Rubber System Compounded with DEA_1_-ESO and its Relationship with Silica-Rubber Morphology

All the samples formulated with DEA_1_-ESO were subjected to strain sweep studies in shear mode at a testing temperature of 100 °C, which was chosen to replicate the thermal environment of tires in use. The reduction in storage modulus as a function of strain (ΔG′ = G′ at 1% strain–G′ at 10% strain) is usually used as a means of quantifying the magnitude of the Payne effect [36,37]. In particular, lower values of ΔG′ are often associated with increases in filler dispersion [38]. The values of ΔG′ (shear mode) for all the samples formulated with DEA_1_-ESO are reported in Table 3. In parallel, the dispersion of silica at the microscale has been investigated by µCT, with the results presented in Table 3. Here, it is important to note that the µCT approach described herein is only able to detect silica agglomerates larger than ~6 µm with precision. Considering the data gathered in Table 3, parallel trends are observed in the ∆G’ and skewness values (first a decrease, then an increase) with increasing modified oil content. A continuous reduction in the equivalent diameter is also observed at all but the highest modified oil concentrations. However, these variations are subtle to the point where their statistical significance is not consistently evident. Overall, then, these results indicate that the observed changes in mechanical properties are not explained by major changes in filler dispersion or structure, and instead arise mainly as a result of increases in crosslink density and/or silica silanization efficiency for the studied compounds. In contrast to the epoxidized soybean oil case [10,11], then, aminated epoxidized soybean oil does not significantly alter the morphology of the silica-filled rubber compound at the micro-scale. The presence of amine groups and their catalytic effects are thus considered to be beneficial in compensating for the drawbacks of epoxidized oils while retaining their beneficial properties as processing aids.

### 3.5. Properties of Silica-Filled Rubber Compound Formulated with Different Aminated Epoxidized Soybean Oils

The data reported in the previous sections demonstrate the capability of DEA_1_-ESO to act as a multipurpose rubber additive, simultaneously enhancing green compound processability and cured compound mechanical properties. At this point, an obvious question arises as to the effect of the amine type on the behavior of the aminated epoxidized soybean oils in the rubber compounds to which they are added. With this in mind, DIPA, DIBA, and bMOEA modified oils were incorporated into the same rubber compound at concentrations of 2.4 vol% (4.5 phr) for comparison purposes. The processability and performance (quasi-static and dynamic mechanical properties) of the resulting compounds were assessed and compared with results from a compound containing 2.4 vol% (4.5 phr) of DEA_1_-ESO and are reported in Table 4. As far as the mixing energy, curing behavior, specific swelling, and the mechanical properties of the compounds containing DEA_1_-ESO, DIPA_1_-ESO, and DIBA_1_-ESO are concerned, there were no statistically significant differences between the materials. This reveals that the bulkiness of the amine group plays no significant role in determining the performance of the resultant aminated epoxidized soybean oil, at least among the amines studied. Previously, it was argued that the alkalinity of the amine controls the alkalinity of the aminated epoxidized soybean oil and its catalytic ability relevant to silica-filled rubber properties. As the reported *pK_a_* values for DEA (*pK_a_* = 11.30 [17]; 10.98 [18]), DIPA (*pK_a_* = 11.05 [18]), and DIBA (*pK_a_* = 10.50 [18]) are all quite similar to one another, then, it follows that the resultant aminated epoxidized soybean oils should possess similar levels of alkalinity and catalytic activity, thus producing similar cured properties in the rubber compounds containing them, as observed in practice. On the other hand, given its much lower *pK_a_* value (*pK_a_* = 8.90 [17]), bMOEA should yield a modified oil (bMOEA_1_-ESO) with significantly reduced alkalinity compared to the previously described dialkylamine-based aminated epoxidized soybean oils, making it less catalytically active and affecting the cured properties of the rubber compound containing it accordingly. Consistent with this argument, the rubber compound containing bMOEA_1_-ESO is found to show a higher scorch time, lower delta torque, and higher values of specific swelling when compared to rubber compounds containing dialkylamine-based aminated epoxidized soybean oils. The increase in specific swelling implies a decrease in crosslink density that is reflected in the quasi-static properties of the rubber compound as a tensile strength equivalent to that of the control formulation. In contrast, all elongation at break and ∆G’ values of the experimental rubber compounds are effectively equivalent to one another and to those of the control formulation. This further confirms that the incorporation of the aminated epoxidized soybean oils does not alter the rubber-filler morphology of silica-filled rubber compounds to a significant degree. This is particularly noteworthy given the fact that bMOEA_1_-ESO is expected to be the most polar oil studied, implying the greatest capacity to interact with the silica surface (as well as the lowest compatibility with the rubber matrix). Nonetheless, it is important to note that all of the modified oils studied here show the capacity to reduce mixing energy vs. the control formulation. This implies that, in spite of the lack of changes in filler dispersion at the microscale, the presence of polar groups enhances processability while the presence of amines in particular (especially more alkaline amines) enhances cured properties. While the alkalinity of the aminated epoxidized soybean oils is found to have no impact on the processability, it appears to be the key parameter for the regulation of the cured properties of the resultant rubber compounds. Taken together, these results indicate that this family of aminated epoxidized soybean oils may justly be described as multipurpose, multifunctional processing additives.

## 4. Conclusions

Aminated epoxidized soybean oils were successfully synthesized using four different secondary amines and then incorporated into a silica-filled model tire tread formulation. When a conventional hydrocarbon process oil was partially replaced by aminated epoxidized soybean oil, a reduction in mixing energy was observed during the NP_1 stage; this was attributed to the increased polarity of these oils and their greater compatibility with the silica filler. Depending on the chemical nature of the grafted amines, the curing rate, the degree of cure, and the crosslink density of the rubber composite were affected as well. Given similar alkalinity, DEA_1_-ESO, DIPA_1_-ESO, and DIBA_1_-ESO accelerated curing and increased crosslink density in an equivalent fashion. This gave increased tensile strength in the cured compounds with no significant change in elongation at break. In contrast, with its lower alkalinity, bMOEA_1_-ESO had less of an influence on curing and cured properties. Based on these results, we conclude that all of the aminated epoxidized vegetable oils studied act to reduce the mixing energy of the rubber compound thanks to the presence of polar functional groups, while changes in cured properties depend on the presence of amine groups with sufficient alkalinity to induce cure acceleration. These materials represent promising candidates as multipurpose, multifunctional processing aids for silica-filled rubber compounds, and provide a convenient means to independently alter processability (through the content of polar groups in general) and cured compound performance (through the content of sufficiently alkaline amines in particular). In addition to their potential utility in existing systems, their mechanism of action is sufficiently general that it may also find utility when added to other elastomeric composites of SBR with all manner of novel multifunctional fillers (see, for instance, [39]).

## 5. Patents

On April 16, 2020, a US provisional patent application was filed in relation to the additives described in this submission and listing Marc Weydert, Frida Nzulu, Arpan Datta Sarma, Pierre Verge, and Daniel Schmidt as inventors. In April of 2021, full patent protection was pursued in the US, the EU, and China.

## Figures and Tables

**Figure 1 polymers-13-03608-f001:**
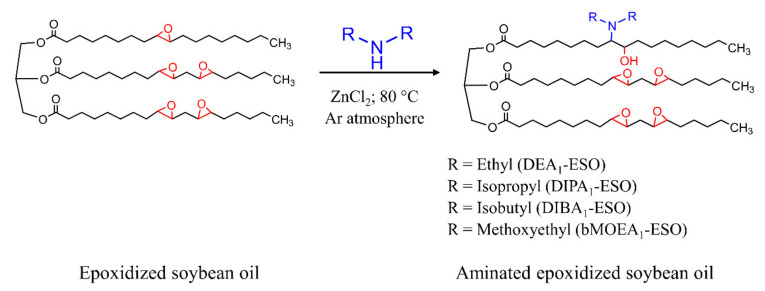
Reaction scheme for amination of epoxidized soybean oil.

**Figure 2 polymers-13-03608-f002:**
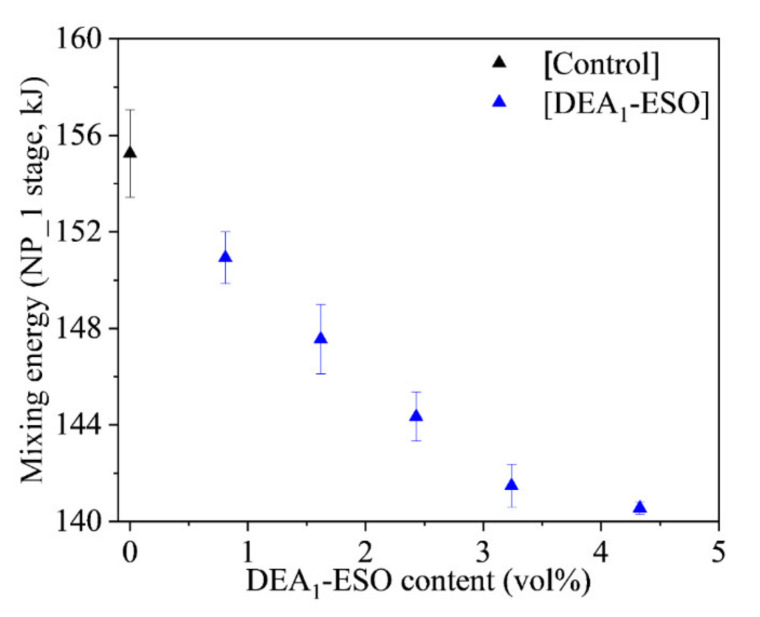
Mixing energy during the NP_1 mixing stage of a series of silica-filled rubber compounds vs. volume percentage of aminated epoxidized soybean oil (DEA_1_-ESO), as compared with the control.

**Figure 3 polymers-13-03608-f003:**
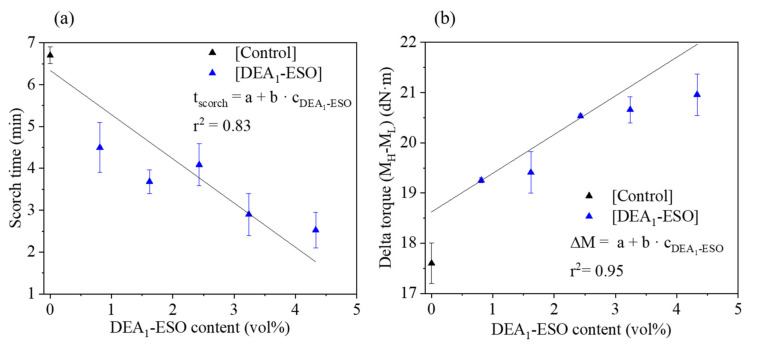
Scorch time (**a**) and delta torque (**b**) values from MDR analyses of silica-filled rubber compounds formulated with DEA_1_-ESO, as compared with the control.

**Figure 4 polymers-13-03608-f004:**
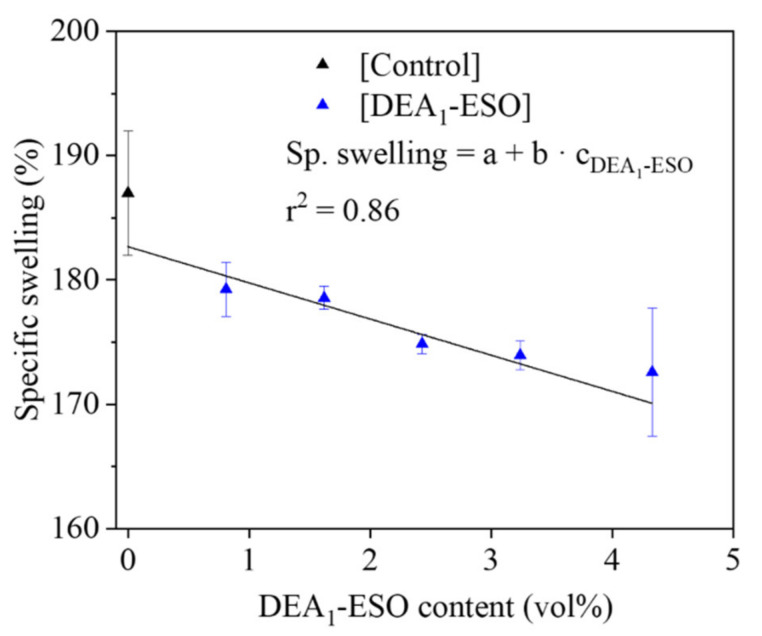
Specific swelling vs. volume percent of aminated epoxidized soybean oil (DEA_1_-ESO) for a series of silica-filled rubber compounds, as compared with the control.

**Figure 5 polymers-13-03608-f005:**
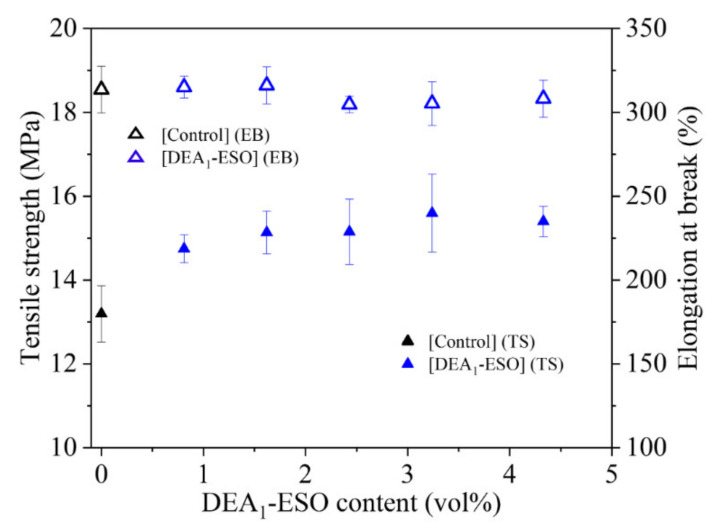
Quasi-static mechanical properties (tensile strength = solid symbols, elongation at break = hollow symbols) of silica-filled rubber compounds vs. volume percent of aminated epoxidized soybean oil (DEA_1_-ESO), as compared with the control.

**Table 1 polymers-13-03608-t001:** Studied formulations of silica-filled rubber compounds formulated with aminated epoxidized soybean oils including their respective volume percentage (vol% = vol. of aminated epoxidized soybean oil × 100/vol. of rubber compound), along with the control formulation; compositions are given in parts per hundred parts of rubber (phr).

Rubber Compound	Experimental Oil		Vol% of A-ESO *^b^*
	*Type*	*Content ^a^* *(phr)*	
[Control]	-	-	0
[DEA_1_-ESO]_1.5–8_	DEA_1_-ESO	1.5/3/4.5/6/8	0.8/1.6/2.4/3.2/4.3
[DIPA_1_-ESO]_4.5_	DIPA_1_-ESO	4.5	2.4
[DIBA_1_-ESO]_4.5_	DIBA_1_-ESO	4.5	2.4
[bMOEA_1_-ESO]_4.5_	bMOEA_1_-ESO	4.5	2.4

Styrene–butadiene rubber (SBR)—80 phr; butadiene rubber (BR)—20 phr; ZnO—2.5 phr; stearic acid—3 phr; silica—80 phr; *N*-(1,3-dimethylbutyl)-*N*′-phenyl-*p*-phenylene diamine (6 PPD)—2.5 phr; bis(3-triethoxysilylpropyl)disulfide (TESPD)—8 phr; 2-mercaptobenzothiazole (MBT)—0.3 phr; diphenyl guanidine (DPG)—3.2 phr; *N*-cyclohexyl-2-benzothiazole sulfonamide (CBS)—2.3 phr; sulfur—1.1 phr. *^a^* The formulations were adjusted to keep the total content of oil constant (26 phr) for all the mix. For instance, the control compound contained 26 phr of TDAE oil, while [DEA_1_-ESO]_4.5_ contained 4.5 phr of DEA_1_-ESO and 21.5 phr of TDAE oil. *^b^* A-ESO refers to aminated epoxidized soybean oil.

**Table 2 polymers-13-03608-t002:** Mixing sequence for all studied rubber compounds.

Non-Productive Stage_1	Non-Productive Stage_2	Productive Stage
Rubber (SBR, BR) + ZnO (0.5 phr) + Stearic acid + Oil (TDAE, modified oil) + Silica (65 phr)	Silica (15 phr) + Silane coupling agent (TESPD) + Anti-oxidant (6-PPD)	Sulfur + Accelerators (MBT, DPG, and CBS)+ ZnO (2 phr)

**Table 3 polymers-13-03608-t003:** Comparison of the Payne effect with morphological parameters including equivalent diameter and skewness of the distribution of equivalent diameters of filler agglomerate for silica-filled rubber compounds formulated with DEA_1_-ESO, along with the control.

Sample Name	Magnitude of Payne Effect (MPa)	Equivalent Diameter (µm)	Skewness
[Control]	0.30 ± 0.01	10.15 ± 0.17	2.77 ± 0.13
[DEA_1_-ESO]_1.5_	0.25 ± 0.02	9.98 ± 0.09	1.89 ± 0.11
[DEA_1_-ESO]_3_	0.27 ± 0.01	9.76 ± 0.19	2.36 ± 0.09
[DEA_1_-ESO]_4.5_	0.32 ± 0.01	8.95 ± 0.07	2.71 ± 0.20
[DEA_1_-ESO]_6_	0.33 ± 0.04	8.65 ± 0.05	2.68 ± 0.16
[DEA_1_-ESO]_8_	0.33 ± 0.01	9.06 ± 0.05	3.03 ± 0.15

**Table 4 polymers-13-03608-t004:** Properties of silica-filled rubber compounds vs. aminated epoxidized soybean oil type of different alkalinity, along with the control.

Sample Name	Mixing Energy (kJ)	Cure Kinetics		Specific Swelling (%)	Quasi-Static Mechanical Properties		Dynamic Mechanical Properties
		∆M (dN·m)	Scorch Time (min)		Tensile Strength (MPa)	Elongation at Break (%)	∆G′ (MPa)
[Control]	155 ± 2	17.6 ± 0.6	6.7 ± 0.2	187.0 ± 5.0	13.2 ± 0.6	313 ± 14	0.30 ± 0.01
[DEA_1_-ESO]_4.5_	144 ± 1	20.5 ± 0.4	4.1 ± 0.5	174.9 ± 0.8	15.2 ± 0.8	304 ± 5	0.32 ± 0.00
[DIPA_1_-ESO]_4.5_	147 ± 1	20.1 ± 1.0	4.9 ± 0.8	173.7 ± 1.3	15.0 ± 0.7	305 ± 10	0.32 ± 0.09
[DIBA_1_-ESO]_4.5_	146 ± 1	19.8 ± 1.0	4.7 ± 0.6	175.0 ± 1.5	15.5 ± 0.3	304 ± 2	0.31 ± 0.05
[bMOEA_1_-ESO]_4.5_	144 ± 1	18.9 ± 0.0	5.7 ± 0.1	185.0 ± 4.0	13.8 ± 0.8	292 ± 13	0.26 ± 0.00

## Data Availability

Not applicable.

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
