# Peer review of "Multipurpose Processing Additives for Silica/Rubber Composites: Synthesis, Characterization, and Application"

_polymers, 2021, doi:10.3390/polym13213608_

Round 1

Reviewer 1 Report

The authors attempt to prepare a new bio-based processing additives, which could promote the mixing of the rubber compound and enhance mechanical properties of rubber composites. This work is novel and interesting. However, there are some issues should be resolved before publishing:

  1. introduction, the authors have better briefly summarized the experiment results at the end.
  2. section 2.3, why two kinds of rubber were used in this research? How about only one kind of rubber?
  3. the content of silica was 80 phr, it was seem to much, is it necessary?
  4. if the control compound contained 26 phr of TDAE oil, the experimental sample should also contain 26 phr of DEA1-ESO, why the sample contain two kinds of oils (4.5 + 21.5)?
  5. the conclusions could be rewriten and should be concise.
  6. Some previously relative works are useful for the author to recognize the progress of this fields and could be cited in this work: Composites Part A, 2021, 145(5): 106292; Composites Science and Technology, 197 (2020) 108231.

Author Response

Reviewer #1:

First, we appreciate that the Reviewer found our work interesting and novel! The text that follows summarizes our efforts to address the issues raised by the Reviewer.

[The Reviewer has asked for clarification as to why the particular rubber formulation used was chosen, and in particular, why it contained two different types of rubber and 80 phr of silica, and why the use of a blend of modified ESO and TDAE oil was studied instead of a system where all TDAE oil was replaced with modified ESO]

We appreciate the question, as we recognize that, as per the Reviewer’s comments, the rubber compounds we worked with were rather complicated in nature. In fact, they were formulated in the context of our partnership with Goodyear Tire and Rubber company, and were designed to be representative of realistic passenger car tire tread compounds in order to ensure the relevance of our results to such materials. A typical passenger car tire tread is generally composed of two or more rubbers (generally SBR and BR, as we have used) to optimize the performance of the tire. We have followed the following references to formulate the recipe:

  1. Dutta, S.; In Science and technology of rubber, 3rd ed; Academic Press: United kingdom, 1978; pp. 529-551.
  2. Braum, M. V.; Jacobi, M. A. M. Silica grafted with epoxidized liquid polybutadienes: Its behavior as filler for tire tread compounds. Rubber Chemistry and Technology 2017, 90 (1), 173–194. https://doi.org/10.5254/rct.16.83760.

Along these lines, we agree with the Reviewer that the silica loading used (80 phr) was rather high. This was also selected in the context of a realistic passenger car tire tread compound, the goal of this work being to show that we could improve the processability of such realistic compounds in spite of these high filler loadings and in response to industrial needs. In spite of the challenges such work presents, we are pleased to report that we succeeded in this regard.

Finally, concerning the process oil content, while we have maintained a loading level of 26 phr in all cases, it is true that we only report on the partial replacement of TDAE oil in this manuscript. While it is true that, from the standpoint of sustainability, it might be more attractive to replace all of the TDAE oil with a bio-based oil, in practice this was avoided for other reasons. First, mixing energy was observed to plateau above a certain level of TDAE replacement, indicating no further benefits to processability if more DEA1-ESO would be added (Figure 2).  Second, the addition of high levels of DEA1-ESO reduces scorch time to the point where process safety might be affected (Figure 3). Finally, as DEA1-ESO is more expensive than TDAE oil, higher loadings of DEA1-ESO are also expected to increase compound cost. For these reasons we focused on the partial replacement of TDAE by DEA1-ESO instead of the complete replacement.

We have edited the manuscript based on the Reviewer’s questions concerning the formulation in use to better clarify these points in the text.

[The Reviewer suggested the addition of a summary of experiments at the end of the introduction and a more concise conclusion section.]

We appreciate the suggestions for strengthening our manuscript and have made edits accordingly. In particular, we now communicate on some of the key results at the end of the introduction, and we have reduced the length of our conclusion section.

[The Reviewer has suggested additional references that might be worth citing in the current manuscript]

Having reviewed the Reviewer’s suggestions, we have now incorporated one of the suggested references (Composites Part A, 2021, 145(5): 106292) into our manuscript.

We thank the Reviewer for their time and effort in reviewing the present article.

Reviewer 2 Report

The authors present a well designed research paper of high interest for the readers from the rubber field. Reading and reviewing of the paper was a pure pleasure. I don't have any critical comments regarding this artice. I recommend the paper for publication.

Author Response

Reviewer #2:

[The Reviewer appreciated the design and representation of the work and has not recommended any changes to the current manuscript, instead suggesting that it be  published as-is]

We thank the Reviewer for their positive feedback, which motivates us to work continuously and constructively in the present domain, and we appreciate their time and effort in this regard!

Round 2

Reviewer 1 Report

It could be accepted.